# Nano targeted Therapies Made of Lipids and Polymers have Promising Strategy for the Treatment of Lung Cancer

**DOI:** 10.3390/ma13235397

**Published:** 2020-11-27

**Authors:** Marwa Labib Essa, Maged Abdeltawab El-Kemary, Eman Mohammed Ebrahem Saied, Stefano Leporatti, Nemany Abdelhamid Nemany Hanafy

**Affiliations:** 1Group of Nanomedicine, Institute of Nanoscience and Nanotechnology, Kafrelsheikh University, KafrElSheikh 33516, Egypt; marwa.essa@nano.kfs.edu.eg (M.L.E.); elkemary@sci.kfs.edu.eg (M.A.E.-K.); 2Pharos University, Alexandria 21648, Egypt; 3Pathology Department, Faculty of Medicine, Kafrelsheikh University, KafrElSheikh 33516, Egypt; eman_saied@med.kfs.edu.eg; 4CNR NANOTEC-Istituto di Nanotecnologia, Via Monteroni, 73100 Lecce, Italy; stefano.leporatti@nanotec.cnr.it

**Keywords:** lung cancer, chemotherapy, nanoparticles, cytotoxicity

## Abstract

The introduction of nanoparticles made of polymers, protein, and lipids as drug delivery systems has led to significant progress in modern medicine. Since the application of nanoparticles in medicine involves the use of biodegradable, nanosized materials to deliver a certain amount of chemotherapeutic agents into a tumor site, this leads to the accumulation of these nanoencapsulated agents in the right region. This strategy minimizes the stress and toxicity generated by chemotherapeutic agents on healthy cells. Therefore, encapsulating chemotherapeutic agents have less cytotoxicity than non-encapsulation ones. The purpose of this review is to address how nanoparticles made of polymers and lipids can successfully be delivered into lung cancer tumors. Lung cancer types and their anatomies are first introduced to provide an overview of the general lung cancer structure. Then, the rationale and strategy applied for the use of nanoparticle biotechnology in cancer therapies are discussed, focusing on pulmonary drug delivery systems made from liposomes, lipid nanoparticles, and polymeric nanoparticles. Many nanoparticles fabricated in the shape of liposomes, lipid nanoparticles, and polymeric nanoparticles are summarized in our review, with a focus on the encapsulated chemotherapeutic molecules, ligand–receptor attachments, and their targets. Afterwards, we highlight the nanoparticles that have demonstrated promising results and have been delivered into clinical trials. Recent clinical trials that were done for successful nanoparticles are summarized in our review.

## 1. Introduction

Lung cancer is the most frequently diagnosed cancer in the world and a common reason for cancer-related deaths [1]. For patients diagnosed with this type of cancer, the 5-year survival rate is approximately 17.8% [2]. Lung cancer can be divided into three main subtypes according to microscopic evidence and histological profiles: non-small cell lung cancer (NSCLC), small cell lung cancer (SCLC), and lung carcinoid tumor, accounting for 85%, 10–15%, and less than 5% of cases, respectively [3]. Among the three subtypes of lung cancer, NSCLC is the most common one diagnosed in non-smokers. It appears in women more than men, and it is more frequently discovered in younger people than other type of lung cancer [4]. This type can be subdivided, according to World Health Organization (WHO), into adenocarcinoma, squamous cell carcinoma, and large cell carcinoma [5].

Adenocarcinoma is a popular type, representing around 40% of all total diagnostic cases. It usually occurs in smokers and nonsmokers [6]. It arises from small airway epithelial cells that form the lining of the lung and alveolar cells: the mucus secreting cells [7]. Furthermore, it grows slowly and can spread outside the lungs. Adenocarcinoma is characterized histologically by the presence of glandular/acinar growth, papillar differentiation, or a single-layer of wallpaper-like spread along the alveolar septum and bronchioles [8]. Squamous-cell carcinoma is derived from the squamous cell type located at the airways of the epithelial cells. These cells line the bronchial tubes in the center of the lungs. This type is mostly associated with smoking tobacco, and it represents 30% of all lung cancer patients [9]. It is histologically identified by the presence of keratinization or intercellular bridges.

Large cell carcinoma comprises 5–10% of lung cancer patients. It arises from the central region of the lungs, the area nearest to the lymph nodes, and the wall of the chest [10]. It usually grows and spreads rapidly, which makes its treatment challenging. Large cell carcinoma, also called non–small cell cancer, has a poor prognosis [11].

SCLC represents 25% of all invasive cancer types worldwide, and it is found exclusively in smokers [12]. It originates from neuroendocrine cell precursors. Thus, it is attributed to endocrine and neurologic paraneoplastic syndromes (Eaton Lambert syndrome, inappropriate antidiuretic hormone secretion, and Cushing’s syndrome) [12]. Moreover, it is characterized by its worse clinical course than that of NSCLC [13,14]. Additionally, it can be resistant to both chemotherapy and radiotherapy courses [15,16].

The last type of lung cancer is lung carcinoid tumor. It originates from neuroendocrine cells, which is are special cells located in the lungs. The growth of this type of cancer is typically very slow and it rarely spreads (see Figure 1) [17].

Here, we try to highlight the development and application of nanoparticles in lung cancer treatment, mainly those made of liposomes, lipids, and polymer materials. Many nanoparticle-based therapies have been developed for the treatment of metastatic NSCLC, such as liposomes, polymeric nanoparticles (NPs), albumin NPs, Lipid NPs, inorganic NPs, and metal NPs. However, few have been translated successfully into clinical trials. In our current study, liposomes, solid lipid nanoparticles (SLNPs), polymeric nanoparticles (PNPs), and hybrid polymeric materials were studied, and they are summarized in Table 1, Table 2 and Table 3, while the successful nanoparticles that moved into clinical trials are outlined in Table 4.

## 2. Nanotechnology-Mediated Pulmonary Drug Delivery

The possible solution for lung cancer treatment involves chemotherapy, surgery, radiotherapy, and the development of targeted nano-therapies [18]. Indeed, many lung cancer patients are not eligible and not applicable for surgery due to diagnosis at a late stage of disease [19]. Clinically, chemotherapies like Cisplatin, Paclitaxel, and Gemcitabine are considered to be the most promising strategies for the treatment of lung cancer [20,21]. However, these chemotherapeutic agents are usually lipophilic in nature and have poor water solubility [22]. For instance, traditional chemotherapies are very aggressive and have huge side effects on normal cells [23]. Thus, there is an urgent need to develop and expand these therapeutics to increase their curing ability and decrease side effects. In this regard, the modification of drug formulas, reduction in the diameters of macromolecules, changes in drug charges, and the possible conjugation of therapeutic agents with biodegradable materials could improve the bioavailability of the drugs [24,25,26].

First, the term nanotechnology can be used to describe the possible control of matter in the size range between 1 and 100 nm. This size range is strongly recommended for biomedical applications [27]. The application of nanotechnology in cancer research has been highly developed in recent decades, providing many advantages, including good stability and the eradication of toxicity [28]. Nanoparticle-based drug delivery systems can be optimized by reducing their diameters, controlling release, and developing better imaging and diagnostic tools for earlier detection of cancer cells in biological systems [29]. Also, a smaller dosage is required for more rapid onset of therapeutic action [30]. Functionalized nanoparticles can further help greatly in the development of targeted therapies by optimizing both active and passive targeted therapies [27]. Passive targeting leads to the enhancement of the permeability and retention effect (EPR), which is defined as a large tumor vasculature and low lymphatic drainage [31].

EPR can increase the distribution of encapsulated cargo molecules, whereas nano-sized particles show good accumulation in tumor tissues because they can rapidly reach a specific location with a higher concentration than other formulations [32]. Meanwhile, active targeting can be developed through the conjugation of drugs to specific ligands that have good affinity to bind receptors that are overexpressed by tumor tissues. Hence, drugs act as prodrugs until they are recognized by the active sites of tumor cells [33] (see Figure 2).

Pulmonary drug delivery systems provide a good opportunity to delivery therapeutic molecules directly into the locations of lung cancer cells, through both systemic and localized treatment forms [34]. The anatomy of the lungs provides potent characterization, enabling nanoparticles to optimize their reactions effectively. Besides that, nanoparticles have been functionalized to have great capacity for solute exchange, due to their good surface area, as well as their ability to bypass the first-pass metabolism [35,36]. In general, nanoparticles represent moieties that could interact with one or more monomers, forming structures that have colloidal suspension properties. According to their size, net charge, chemical structure, and functionalization, they can be classified as either having toxicity or being safe for biomedical application. The toxicity of nanoparticles depends mainly on their biodegradability and cross-links [37]. For this reason, scientists have given much attention to this challenge, and now, many biodegradable nanoparticles have been fabricated, and some of them have already been added to clinical trials and sold in the industrial market [38].

For this reason, targeted nano-carriers are used extensively to localize chemotherapeutic agents inside the lungs. This decreases the effect of systemic dilution by increasing the chance of their accumulation within the tumor site as well as reducing the cytotoxicity of chemotherapeutic agent [39]. This has improved patient compliance and upgraded their quality of life [40]. However, their proper delivery faces several challenges, including enhancement of the pulmonary clearance mechanism with an increasing nanoparticle diameter [41,42].

It has been reported that micro and macro particles are deposited at the oropharynx and upper respiratory tract via impaction, while nano-sized particles can reach the site of action easily. Therefore, the particle size range must be carefully controlled. In addition, many pulmonary clearance mechanisms have been observed, such as alveolar macrophages and the mucociliary escalator system, which prevents the adsorption and stabilization of nanoparticles. The mucociliary system can remove most of the deposited insoluble particles with sizes > 6 mm, and most of them can be phagocytized [43]. Various nanocarriers have obtained great advantages in diverting the drugs into aerosols and providing highly sustainable nebulization forces, and these have been used recently in the treatment of lung cancer [44]. For this reason, many nanoparticles have been designed and developed for the treatment of lung cancer, such as liposomes, micelles, solid lipid nanoparticles, and polymeric nanoparticles [45]. Indeed, lipid-based NPs and polymeric nanoparticles (PNPs) are considered to be successful delivery systems. These assemblies have been accepted by the United States Food and Drug Administration (FDA) [46]. Lipid-based nanocarriers can be further classified into liposomes, solid lipid nanoparticles, and hybrid polymeric lipid nanoparticles.

### 2.1. Liposomes

Liposomes are phospholipid bilayer spherical systems composed of a core surrounded by a shell [47]. They are formed by a spontaneous vesicle that is separated by water compartments. Phospholipids contain a hydrophilic head that is exposed outwards, while the tail is moved into the bilayer as the hydrophobic stage (Figure 3). Their diameters mostly range from around 20 nm to several micrometers based on the growth of the bilayer’s assembly [48]. Liposomes have many properties and unique varieties that make them the most widely used as drug-delivery systems. They have great potential application for the encapsulation of chemotherapeutic agents, owing to their ability to increase the solubility and permeability of these agents as well as allowing the encapsulation of both hydrophilic and hydrophobic cargo molecules. Additionally, they are biocompatible and biodegradable materials with non-immunogenicity [49]. Increasingly, the possible modification of liposomes by polyethylene glycol (PEG) is improving their stability and therapeutic activity, whereas, it has led to the minimization of their clearance [50,51].

The PEGylated process can cause changes in the physicochemical properties of nanoparticles, resulting in changes to their hydrophilicity, diameter, conformation, and intermolecular interactions [52]. Both PEGylated and conventional liposomes are able to deliver their encapsulated drugs through passive targeting delivery [53]. Moreover, the further modification of liposomes by specific ligand conjugations can increase the selectivity of liposome-mediated delivery towards lung cancer cells [54]. Many strategies involving the targeting of overexpressed surface receptors or specific molecules, like epidermal growth factor receptor, sigma receptors, folate receptors, peptides, and hyaluronic acid, have appeared recently. Sometimes, they have been designed to drive directly towards specific cell organelles (e.g., mitochondria). In other cases, they can target the tumor microenvironment (through integrins, cluster of differentiation 44 (CD44) or vascular cell adhesion molecules) [55].

Koshkina and her coworkers used 9-nitrocamptothecin (9NC)-loaded liposomes as an inhalation treatment against lung cancer, leading to reduced lung weight and minimizing the number of tumor foci in two different lung metastasis models [56]. In the same way, a doxorubicin (DOX)-loaded PEGylated liposomal formulation has been approved for cancer therapy recently [57]. DOX has expressed its ability to provide significant results in the treatment of SCLC and NSCLC patients. Thus, it can be used in the single state or combined with other conventional therapies [58]. Similarly, Miatmoko and his coworkers reported that poly L glutamic acid (PGA) could be used as a stabilizer for encapsulated DOX, leading to the accumulation of DOX in Lewis lung carcinoma (LLC) tumors [59]. Ierano and his coworkers delivered peptide R and DOX as a combined therapy for metastatic melanoma lung cancer, since peptide R can inhibit chemokine receptor (CXCR4). In this case, these combinations were found to have a synergic effect against CXCR4 expression, leading to the reduction of lung metastases [60,61].

Table 1 shows that Tian and his team [62] designed hyaluronic acid (HA)-modified mitochondria targeting smart liposomes with PTX to overcome multidrug resistance (MDR). The liposomes were prepared by thin-film hydration and then coated with hyaluronic acid (HA) by electrostatic adsorption, obtaining particle sizes of about 100–140 nm. The results showed that the accumulation of nanoparticles in the mitochondria of tumor cells causes significant inhibition of cancer proliferation, leading to MDR being overcome.

**Table 1 materials-13-05397-t001:** Nano-liposome encapsulating different types of drugs for the treatment of lung cancer.

Drug	Composition	Preparation Technique	Target	Ref.
Paclitaxel (PTX)	A PTX liposome coated with hyaluronic acid (HA) that acts as an active drug targeting the mitochondria to overcome multidrug resistance.	Thin-film hydration	Adenocarcinomic human alveolar basal epithelial cells and taxol cells (Mitochondria)	[62]
PTX	The specific ligand peptide (T7) and the cationic cell-penetrating peptide TAT were connected with phospholipids via a polyethylene glycol (PEG) spacer to prepare dual-ligand liposomes	Thin-film hydration methods	Adenocarcinomic human alveolar basal epithelial cells (transferrin receptor)	[63]
Afatinib (AFT)	pH-sensitive liposomes for tyrosine kinase inhibitor AFT	Thin-film. Hydration followed by extrusion	Non-small cell lung cancer (NSCLC)	[64]
Docetaxel (DTX)	A15 cell surface modified DTX liposome	Thin-film hydration	Lung cancer stem cells (CD133)	[65]
DTX	Hyaluronic acid (HA)-coated PLGA nano-particulate DTX	Modified nano-precipitation method	Non-small cell lung cancer (NSCLC) (CD44)	[66]
DTX	DTX-loaded folic-acid-conjugated liposomes	Thin-film hydration and ultrasonic dispersion technology	Non-small cell lung cancer (NSCLC) (Folate receptor)	[67]
Erlotinib	Multifunctional liposomal complexes, where anti- epidermal growth factor receptor (EGFR) Apt-conjugated chitosan (Apt-Cs) was anchored into liposomes to co-administrate erlotinib and perfluorooctylbromide (PFOB) for the reversal of hypoxia-induced drug resistance	Thin-film hydration	Non-small cell lung cancer (NSCLC) (EGFR)	[68]
Doxorubicin (DOX)	DOX-loaded thermo-sensitive hybrid liposome formulation composed of dipalmitoyl phosphatidyl cholineas the phospholipid and Poloxamer 188	Thin-film hydration flow by the remote-loading method	Lung cancer (PLA2)	[69]
Pixantrone (PIX)	Poly (sialic acid)–octa decylamine conjugate was synthesized and used to decorate the surface of pixantrone loaded liposomes	Remote loading method	Non-small cell lung cancer (NSCLC)	[70]

Additionally, a dual target therapy was developed by Wang and his team using the specific ligand peptide HAIYPRH (T7) and the cationic cell-penetrating peptide TAT that were connected with phospholipids via a polyethylene glycol (PEG) spacer to prepare the dual-ligand liposomes (T7/TAT-LP-PTX). This structure was evaluated on the tumor spheroids, which revealed that T7/TAT-LP was more efficaciously internalized in tumor cells than TAT-LP, T7-LP, and LP, respectively [63]. Also, Almurshedi and his team [64], studied the preparation of cationic (CL) and pH-sensitive liposomes (PSL) for the tyrosine kinase inhibitor afatinib (AFT), which was developed to enhance the tumor-targetability against NSCLC cells in comparison with conventional liposomes (NL). The obtained liposomes showed a particle size of ˂100 nm with a spherical shape. PSL, CL, and NL showed slow release profiles at pH 7.4. However, at acidic pH values, PSL exhibited fast release, which improved its tumor targetability. Moreover, AFT-loaded PSL inhibited the cell growth of human lung cancer cells (H-1975) more efficiently than free AFT, CL, or NL based on the Annexin V assay.

Zhu and his coworkers [67] investigated the efficacy of pulmonary delivery using docetaxel–folic-acid-loaded liposomes (LPs-DTX-FA) as anti-cancer drugs in the form of nanoparticulate dry powders. It has been suggested that liposome-based inhaled dry powders represent a possible clinical therapy by achieving higher anti-cancer activity at tumor sites and causing less damage to healthy organs.

Tagami and his team incorporated Poloxamer 188 (P188) into liposome-DOX membranes. The results suggested that DOX-loaded DPPC/P188 liposomes may be useful for treating lung cancer [69].

Zhang and his team developed poly(sialic acid)-conjugated octadecylamine, and then the mixture was attached to liposomal pixantrone. The structure represents a promising approach for lung cancer treatment [70]. Many chemotherapeutic molecules have been encapsulated inside moieties of liposomes to target lung cancer, including paclitaxel (PTX), afatinib (AFT), docetaxel (DTX) erlotinib, doxorubicin (DOX), and pixantrone (PIX). These therapeutic molecules are summarized in Table 1.

### 2.2. Solid Lipid Nanoparticle (SLNs)

SLNs are formed by the replacement of the liquid lipid core with a solid one. They are generally solid at 37 °C with diameters of less than 80 nm after filtration [71]. SLNs have good biochemical properties, including low toxicity, controlled drug release, easy modification, small diameters, which facilitates their penetration, and improved physical stability, because there is no hydrolyzed lipid core in the aqueous solution [72]. They are usually synthesized by high-pressure homogenization, double emulsion, high-shear homogenization, and emulsifier evaporation. In this case, cargo molecules can be incorporated in lipid moieties [73,74]. Their diameters can grow during fabrication, forming micro and macro particles [75], and their crystalline structure minimizes their incorporation rate. However, they are used widely as drug carriers for lung cancer therapy (Figure 4) [76].

Videirae and his team evaluated the use of paclitaxel inserted into lipid nanocarriers to determine their therapeutic efficiency in lung cancer. It was reported that these nanocarriers reduced the number and size of lung metastases more in comparison to intravenous injection of the same non-encapsulating drugs [77]. Additionally, Hu and his coworker evaluated the insertion of epirubicin (EPI) into solid lipid NPs as an effective lung cancer therapy. That study confirmed that encapsulated epirubicin inside lipid NPs has greater efficiency than the administration of non-encapsulated EPI solution [78]. Levet and his coworkers produced cisplatin microcrystals loaded inside solid lipid NPs, and these caused deep lung depositions [79].

Drugs that have been encapsulated inside SLNPs are summarized in Table 2. Gupta et al. developed solid lipid core nanocapsules (SLCN). This structure comprises a solid lipid core and a PEGylated polymeric corona. Paclitaxel (PTX) and erlotinib (ERL) have been encapsulated inside this structure and delivered into non-small cell lung cancers [80].

Kabary et al. designed layer-by-layer alternate adsorption involving the use of lactoferrin (LF) and hyaluronic acid (HA) on the surface of lipid nanoparticles (NPs) for dual delivery of berberine (BER) and rapamycin (RAP) to lung cancer. The obtained results showed a high recommendation for the use of LbL for enhanced functionalization of NPs and cellular uptake [81].

Khatri et al. optimized Artemether-loaded solid lipid nanoparticles to treat lung cancer. The results exhibited an anti-lipolytic effect against lung cancer [82].

Rosière, et al. designed solid lipid nanoparticles coated with folate-grafted polyethylene glycol (PEG) and chitosan. Then, paclitaxel was loaded into the SLN. The results showed a positive impact of the coated SLN on the delivery of paclitaxel by inhalation [83].

Soni, et al. developed solid mannosylated lipid nanoparticles loaded with gemcitabine using emulsification and solvent evaporation process. The result was improved cellular uptake and drug efficacy of gemcitabine inside lung cancer [84].

**Table 2 materials-13-05397-t002:** SLNPs encapsulating different types of drugs for the treatment of lung cancer.

Drug	Composition	Preparation Techniques	Target	Ref.
Paclitaxel (PTX) and Erlotinib (ERL)	PTX- and ERL-co-loaded SLCN consisting of cores that accommodate both PTX and ERL and block copolymer coronae with PEGylated exterior.	Nanoprecipitation and sonication	Non-small cell lung carcinoma NSCLC (patients with activating-EGFR mutations)	[80]
Bberberine (BER) and Rapamycin (RAP)	Lactoferrin and HA were utilized to develop layer-by-layer lipid nanoparticles (NPs) for the dual delivery of BER and RAP to lung cancer.	Hot homogenization method	Adenocarcinomic human alveolar basal epithelial cells (HA targeted CD44 receptor)	[81]
Artemether (ART)	Oral anticancer drug ART-SLNs stabilized using MPEG2000- 1,2-distearoyl-sn-glycero-3-phosphoethanolamine-N-[amino(polyethylene glycol)-2000]-N-(Cyanine 5 (,DSPE) exhibited an anti-lipolytic effect	Pressure homogenization technique	Lung cancer	[82]
PTX	Inhaled drug delivery system. Paclitaxel-loaded SLN folate-grafted copolymer of PEG and N-((2-hydroxy-3-trimethylammonium)propyl) chitosan chloride coated liposomes. Surface modification was conducted to prolong the retention of PTX within the lungs	Carbodiimide-mediated coupling chemistry	Lung cancer (Folate receptors)	[83]
Gemcitabine (GmcH)	GmcH loaded mannosylated SLNs for improving drug uptake into the lung cancer cells	Emulsification and solvent	Lung cancer (Mannose receptor)	[84]
Erlotinib (ETB)	ETB-loaded SLN-based formulation of inhaled dry powder	Hot homogenization method followed by sonication	Non-small cell lung carcinoma (NSCLC)	[85]
PTX	PTX- and DNA-loaded NLC was prepared, and surface modification was done by transferring receptor -containing ligands.	Micro-emulsion technique.	Lung adenocarcinoma cell (Transferring receptor)	[86]
PTX and DOX	Combined delivery of PTX and DOX was prepared as a synergistic anti-tumor drug	Melted ultrasonic dispersion method	Non-small cell lung carcinoma (NSCLC)	[87]

Shao, et al. developed transferrin (Tf)-decorated nanostructured lipid carriers as a multifunctional nanomedicine for the co-delivery of paclitaxel (PTX) and enhanced green fluorescence protein plasmid [86].

Wang, et al. developed a method involving combined delivery of PTX and DOX using a melt-emulsification technique with nanostructured lipid carriers. The results showed a highly cytotoxic effect for all formulations in vitro, as compared to single drug delivery [87].

### 2.3. Polymeric Nanoparticles (PNPs)

PNPs are produced in the final colloidal nanoparticle suspension, and they can be synthesized by polymer self-assembly [88]. Good results have been obtained regarding the release of the anti-neoplastic drug into the lungs in a control manner, since they enhance the therapeutic efficacy [89]. Various polymers can be used in the preparation of PNPs, such as polylactic acid (PLA), chitosan (CHI), poly lactic-co-glycolic acid (PLGA), and poly alkyl cyanoacrylates (PAC). FDA has approved PEG and PLGA polymers for potential therapeutic applications that are more effective against carcinomas [90].

Yordanov and his coworkers fabricated epirubicin-loaded poly butyl cyanoacrylate nanoparticles in an aqueous dispersion. They were assembled with Pluronic F68 and Polysorbate 80, as non-ionic surfactants. Fluorescent imaging done in the adenocarcinoma cells showed that the free cargo molecules were highly internalized into the cell nucleus, whereas the drug loaded nanoparticles were accumulated in the cytoplasm. This may explain why cellular internalization was done by endocytosis. The final result suggested that nanoparticle-based anthracycline should be developed as a therapeutic tool for lung adenocarcinoma [91].

Polymeric nanoparticles can be formed by self-assembly or they can be conjugated with proteins like albumin or gelatin as hybrid polymeric protein nanocarriers (HPPNCs). In this special case, the platform takes advantage of both proteins and polymers for optimizing drug carriers. These hybrid materials are used to encapsulate genetic materials and naturally derived vectors, such as viral vectors [92]. Using the same strategy, polymeric hybrid lipid nanoparticles (PHLNPs) are considered highly attractive nanoplatforms for designing anticancer carriers (Figure 5). They are prepared by conjugation of the core and shell of a polymer (which has biodegradable property) and liposome, since this core can cause physical stability and structural integrity, while the shell can facilitate biological internalization. Both hydrophobic and hydrophilic cargo molecules can be integrated inside PHLNP moieties. The hydrophilic drugs can be encapsulated inside the core, while hydrophobic drugs can be entrapped in the shell [93].

Hitzman and his coworkers fabricated PHLNPs with sizes ranging from 400 to 600 nm. They were co-formulated with the presence of polyglutamic acid in the core, while 5-fluorouracil was doped in the shell. In this case, the hydrophobic structure of the lipid shell was assembled by the spray drying technique in the presence of tripalmitin/cetyl alcohol, resulting in nanoparticles with a diameter of 0.9–1.2 mm [94]. The drug release from this capsule occurred in a controlled manner compared to that of different types of liposome and polymer self-assembly. The high viscosity of polyglutamic acid plays a critical role in drug release. For this reason, the diameter of nanoparticles and the lipid shell thickness are important factors in the control of drug release from PHLNPs. There is a relationship between the thickness of the shell of PHLNPs and drug dissolution, because the shell allows the entry of H_2_O. For this reason, a thickness reduction of the shell from 300 to 100 nm can cause rapid drug release from 79% to 85% in 24 h. In other assemblies, the shell of PHLNPs can be formed by polymers and lipids included as a core. This assembly can minimize the clearance rate from the lung [93].

Similarly, paclitaxel-loaded hybrid polyethylene glycol and 2 distearoyl phosphatidylethanolamine micelles reached the deep lung tissues via nebulization [95]. Using this type of assembly, micelles cannot be recognized by macrophages. Additionally, these NPs accumulate in the lung. Moreover, NPs have been observed to have a 45-fold higher area under the curve in rat lungs than intravenous administration.

The previous methods used for the fabrication of polymeric nanoparticles are summarized in Table 3. Menon and his team developed a folate receptor nanoparticle containing a poly(N-isopropylacrylamide)-carboxymethyl chitosan shell and poly lactic-co-glycolic acid (PLGA) core. This assembly enhanced localized chemo-radiotherapy treatment for lung cancers [96].

Manadal and his team developed lipid-monolayer-coated biodegradable polymeric cores (CSLPHNPs) for the delivery of erlotinib to treat non-small cell lung cancer. This assembly was produced by a single-step sonication method using polycaprolactone (PCL) as the biodegradable polymeric core and a phospholipid shell composed of hydrogenated soy phosphatidylcholine (HSPC) and 1,2-distearoyl-sn-glycero-3-phosphoethanolamine-N-[methoxy(polyethylene glycol)-2000] (DSPE-PEG2000). The structure obtained particle sizes of about 170 nm, Polydispersity index(PDI) of <0.2, and a drug entrapment efficiency of about 66% with good serum and storage stability [97].

Xiong and his coworkers designed a combined therapy using cisplatin and metformin. Cisplatin was conjugated to polyglutamic acid to form an anionic mixture, and then the mixture was reacted with cationic polymeric metformin. The payload was then inserted into a liposome composed of DOTAP(2,3-Dioleoyloxy-propyl)-trimethylammonium/Cholesterol/DSPE-PEG-anisamide aminoethyl.

Cisplatin-loaded nanoparticles exhibited significantly increased tumor accumulation compared to free ones without causing any nephrotoxicity [98].

Mottaghitalab and her team optimized SP5-52 peptide conjugated silk fibroin nanoparticles loaded with gemcitabine to treat induced lung tumors in a mice model. The results showed a significant reduction in lung tumor size compared with the untreated groups [99].

**Table 3 materials-13-05397-t003:** Polymeric nanoparticles encapsulating different types of drugs for the treatment of lung cancer.

Drug	Composition	Preparation Technique	Target	Ref.
NU7441—radiosensitizer and Gemcitabine	A folate receptor targeting multifunctional, dual, drugs-loaded nanoparticles containing a poly(N-isopropylacrylamide)-carboxymethyl chitosan shell and a PLGA core to enhance localized chemo-radiotherapy	Standard emulsion method	Human dermal fibroblasts (HDFs) and Alveolar Type 1 epithelial cells	[96]
Erlotinib	Erlotinib-loaded core-shell-type lipid–polymer hybrid nanoparticles composed of polycaprolactone as the core and hydrogenated soy phosphatidylcholine/1,2-distearoyl-sn-glycero-3-phosphoethanolamine-N-(methoxyPEG2000) as the shell	Single-step sonication method	Non-small cell lung cancer (NSCLC).	[97]
Cisplatin (CDDP) and Metformin	Co-encapsulation of CDDP and metformin into single self-assembled core-membrane NPs. CDDP was first conjugated to PGA to form PGA-CDDP, which was electrostatically complexed with the cationic polymer metformin (polymet) and then coated with PEGylatedcationic liposomes to form the final core–membrane structure.		Non-small cell lung cancer (NSCLC).	[98]
Gemcitabine (Gem)	Silk fibroin nanoparticles (SFNPs) used for the systemic delivery of gemcitabine (Gem) For targeting the tumorigenic lung tissue, SP5-52 peptide was conjugated to Gem-loaded SFNPs.	Electrospraying and desolvation method	Animal lung cancer	[99]
paclitaxel (PTX)	PTX-loaded polymeric nanoparticles (PTX-NPs) combined with circadian chronomodulated chemotherapy The polymer nanoparticles were prepared from two amphiphilic three-block copolymers: poly (ε-caprolactone)-poly (ethylene glycol)-poly (ε-caprolactone)	Thin film dispersion technique	Non-small cell lung cancer (NSCLC).	[100]

As a comparison between liposomes, solid lipid nanoparticles and hybrid polymeric lipid nanoparticles, it can be summarized that liposomes are mostly coformulated by the presence of a phospholipid and cholesterol core and shell in their structure [101]. Their diameters depend mainly on the method of fabrication, surfactants, concentrations of phospholipids and cholesterol, dehydration time, and temperature. The architecture of liposomes has obtained much interest in terms of encapsulating hydrophilic drugs inside the core and hydrophobic drugs inside the shell [102]. Hence, this assembly is characterized by the encapsulation of more than one type of drug with different physicochemical properties. Additionally, the spherical bilayer system causes the barrier to prevent the leakage of drugs out of liposome moieties [103].

In this way, liposomes have great advantages as biodegradable and biocompatible materials with the potential for large-scale production during fabrication, allowing them to be used widely in biomedical applications. Besides that, entrapped drugs can be modulated inside the core and shell.

The disadvantage of liposomes is mostly attributed to the possible oxidation of the liposomal phospholipid layer, leading to hydrolysis of the membrane and degradation of the liposomal structure. This drawback leads to the leakage of drugs out of the liposome. Furthermore, lipid peroxidation leads to damage to the properties of the lipid structure, particularly cellular permeability. Additionally, temperature, pH, and light may induce instability into the physical and chemical properties of liposomes, leading to a reduction in the shelf life of liposomes during long-term storage. The other disadvantages could be related to the use of the freeze-drying technique during liposomal fabrication [104], since this technique leads to the rupture of phospholipid membranes. To overcome these drawbacks, liposomes are mostly coated with polymeric materials, such as a PEGylated layer, during fabrication or incorporated inside CaCO3 using the layer by layer technique.

Solid lipid nanoparticles are just a solid core of lipids formed by one of the following techniques at 37 °C: high-pressure homogenization, double emulsion, high-shear homogenization, or emulsifier evaporation [105]. The lipid core can be then functionalized using a PEGylated layer or by alternate adsorption [106]. Solid lipid nanoparticles either have one drug incorporated inside their core during fabrication, or they may contain more than one drug [107]. In general, both hydrophobic and hydrophilic drugs can be inserted. For instance, solid lipid nanoparticles are characterized by biocompatibility, low toxicity, good stability, and enhancement of entrapped lipophilic drugs.

The disadvantage of this structure is mostly associated with the type of lipid used, as this can produce cytotoxicity related to free fatty acids and can cause a growth in the diameter during fabrication [108]. Additionally, the diameter grows during the fabrication of micro- and macroparticles [109].

Polymeric nanoparticles can be identified as an interaction of two opposite polymeric materials that form a cross-linked network and condensed core [110]. This core can be further functionalized by a PEGylated layer or coated by a lipid monolayer. Hanafy et al. optimized hybrid polymeric lipid nanoparticles by using chitosan-oleic acid after blocking its free fatty acid [111]. This strategy reduced the cytotoxicity that can be generated by oleic acid. Hybrid polymeric lipid nanoparticles can be co formulated in the shape of micelles by using polymer self-assembly, while drugs can either be attached during fabrication or inserted after fabrication. Recently, polymeric materials in the shape of hydrogel materials, mucoadhesive materials, and stimuli-responsive polymers have been developed. Polymeric nanoparticles are characterized by controlled drug release, stability inside cells, and easy and cost-effective formulation. However, the disadvantage of polymeric NPs is mostly associated with the type of organic solvent used during fabrication and the polymer cytotoxicity.

Recently, advanced nanotechnology has undergone significant progression by using materials for the production and design of an optimal drug delivery system, since the physicochemical properties of polymers are used as an advantage to develop both liposomes and solid lipid NPs and to overcome their drawbacks during fabrication. These new structures are called polymeric hybrid lipid nanoparticles. Polymeric hybrid lipid nanoparticles contain three main structures, as follows: (1) a hydrophobic/hydrophilic polymeric core inserted inside liposomes or coated by a lipid monolayer; (2) a solid lipid core or liposomal phospholipid/cholesterol layer surrounded by a polymeric shell; (3) polymer materials incorporated inside solid lipid or liposomal phospholipid/cholesterol layers and then and an outer component consisting of a PEGylation.

## 3. Clinical Studies of Pulmonary Nanoparticles Can Be Finally Driven into the Industry Market

Clinical trials represent strategy-based scientific research that attempts to evaluate the influence of a new therapeutic molecule on human health outcomes. It includes several steps, as follows: Phase I—the study of new therapeutic molecules on a small group of volunteers; Phase II—monitoring and evaluation of safe drugs on a large group of volunteers; Phase III—the study of safe drugs in different groups of volunteers located in different regions and countries; and Phase IV—monitoring of safe drugs after obtaining approval for their use in a wide population over long time frame [112]. During the clinical trial process, each drug is given a particular description according to its status, as summarized in Table 4. For instance, it is “recruiting” when several participants are invited to contribute to the study, while “active” status means the study is underway. On the other hand, when the investigation is finished, there is no need for any participants. The status of the drug is changed to “completed”. “Terminated” status means that the study has stopped and will not be started again [113]. In the current review, we have described the use of several nanochemotherapeutic agents to treat lung cancer, and recently, they have reached clinical trial status. Irinotecan liposome injection has obtained a “recurring” status and is being investigated in comparison with topotecan in patients with small cell lung cancer after treatment with a platinum-based first-line therapy (Phase 3; 2018–2022 study). Many studies have completed the clinical study process, such as Cholesterol-Fus1 in non-small-cell lung cancer [114] and the liposomal form of Lurtotecan as OSI-211 to treat recurrent small cell lung cancer [115]. Stimulating Targeted Antigenic Responses To NSCLC (START) was a phase III trial of the MUC1-antigen-specific cancer immunotherapy tecemotide, following chemoradiotherapy for unresectable stage III NSCLC [116] and a liposomal formed of Lurtotecan [7(4-methylpiperazinomethylene)-10,11-ethylenedioxy-20-(S)-camptothecin dihydrochloride] was combined with cisplatin to treat patients with advanced or metastatic solid tumors. A phase II trial of this combination showed that three of 25 patients with breast cancer and two of 23 patients with NSCLC had partial responses [117].

**Table 4 materials-13-05397-t004:** Clinical trial studies of lung cancer therapies using lipid nanoparticles.

Study Title	Conditions	Status	Assigned Number
Study of Irinotecan Liposome Injection (ONIVYDE^®^) in Patients With Small Cell Lung Cancer	Small Cell Lung Cancer	Recruiting	NCT03088813
Irinotecan Hydrochloride Liposome Injection (LY01610) For Small Cell Lung Cancer	Small Cell Lung Cancer	Recruiting	NCT04381910
Paclitaxel Liposome for Squamous Non-Small-cell Lung Cancer Study (LIPUSU)	Squamous Non-small-cell Lung Cancer	Active, not recruiting	NCT04381910
Phase I Study of IV DOTAP: Cholesterol-Fus1 in Non-Small-Cell Lung Cancer	Lung Cancer	Completed	NCT00059605
BLP25 Liposome Vaccine and Bevacizumab After Chemotherapy and Radiation Therapy in Treating Patients With Newly Diagnosed Stage IIIA or Stage IIIB Non-Small Cell Lung Cancer That Cannot Be Removed by Surgery	Lung Cancer	Active, not recruiting	NCT00828009
Efficacy and Safety Study of OSI-211 (Liposomal Lurtotecan) to Treat Recurrent Small Cell Lung Cancer	SCLC and Carcinoma, Small Cell	Completed	NCT00046787
Study of Tecemotide (L-BLP25) in Participants With Stage III Unresectable Non-Small Cell Lung Cancer (NSCLC) Following Primary Chemoradiotherapy	Non-small Cell Lung Cancer	Completed	NCT00960115
Liposomal Lurtotecan Plus Cisplatin in Treating Patients With Advanced or Metastatic Solid Tumors	Head and Neck Cancer, Lung Cancer, Ovarian Cancer	Completed	NCT00006036
Study of Autologous CIK Cell Immunotherapy Combination With PD-1 Inhibitor and Chemotherapy in Advanced NSCLC	Non-small Cell Lung Cancer	Recruiting	NCT03987867
A Study of FF-10850 Topotecan Liposome Injection in Advanced Solid Tumors	Advanced Solid Tumors	Recruiting	NCT04047251
TUSC2-Nanoparticles and Erlotinib in Stage IV Lung Cancer	Lung Cancer	Active, not recruiting	NCT01455389
Doxil Topotecan, Doublet Cancer Study	Small Cell Lung Cancer, Pancreatic Cancer, Head and Neck Cancer	Completed	NCT00252889
TUSC2-nanoparticles (GPX-001) and Osimertinib in Patients With Stage IV Lung Cancer Who Progressed on Osimertinib Alone	Carcinoma, Non-Small-Cell Lung	Not yet recruiting	NCT04486833
Intrathecal Pemetrexed for Recurrent Leptomeningeal Metastases From Non-Small Cell Lung Cancer	Leptomeningeal Metastases	Completed	NCT03101579
Inhaled Doxorubicin in Treating Patients With Primary Lung Cancer or Lung Metastases	Lung Cancer, Metastatic Cancer	Completed	NCT00004930
VX-710, Doxorubicin, and Vincristine for the Treatment of Patients With Recurrent Small Cell Lung Cancer	Lung Cancer	Terminated	NCT00003847
Topotecan Hydrochloride and Doxorubicin Hydrochloride in Treating Patients With Relapsed or Refractory Small Cell Lung Cancer	Recurrent Small Cell Lung Carcinoma	Completed	NCT00856037
Inhaled Doxorubicin in Treating Patients With Advanced Solid Tumors Affecting the Lungs	Lung Cancer, Malignant Mesothelioma, Metastatic Cancer	Completed	NCT00020124
Effects of STM 434 Alone or in Combination With Liposomal Doxorubicin in Patients With Ovarian Cancer or Other Advanced Solid Tumors	Ovarian Cancer, Fallopian Tube Cancer, Endometrial Cancer, Solid Tumors	Completed	NCT02262455
An Open-label, Phase I/IIa, Dose Escalating Study of 2B3-101 in Patients With Solid Tumors and Brain Metastases or Recurrent Malignant Glioma.	Brain Metastases, Lung Cancer Breast Cancer	Completed	NCT01386580
Radiation Therapy Plus Combination Chemotherapy In Treating Patients With Limited Stage Small Cell Lung Cancer	Lung Cancer	Completed	NCT00003364
A Phase II Study of Doxorubicin, Cyclophosphamide and Vindesine With Valproic Acid in Patients With Refractory or Relapsing Small Cell Lung Cancer After Platinum Derivatives and Etoposide	Small Cell Lung Carcinoma	Completed	NCT00759824
Combination Chemotherapy Followed by Radiation Therapy in Patients With Small Cell Lung Cancer	Lung Cancer	Completed	NCT00002822

Data retrieved from the US National Institutes of Health website (http://clinicaltrials.gov/) on 21 August 2011 [113].

## 4. Conclusions

Nanotechnology has been used recently as an interesting approach to develop cancer therapies [118,119,120,121,122,123,124]. Functionalized nanoparticles have an interesting role in targeting lung cancer cells by driving an optimum dose into a specific reaction using ligand–receptor conjugation. Liposomes, solid lipid nanoparticles, and polymeric-based nanoparticles are able to overcome the respiratory tract barriers and mucociliary clearance to deliver drugs into the deep part of the lungs. Although many type of nanoparticles have been fabricated recently, liposomes have potential application in the biomedical field and are being entered into clinical trials because of their biodegradability, biocompatibility, their ability to undergo large-scale production, and their ability to be functionalized by PEGylation or by using layer-by-layer assembly, allowing prolongation of the circulation half-life. Additionally, they have a small diameter, low toxicity, hybrid structure, the ability to provide controlled and sustained release, and the ability to modulate the distribution of drugs inside the core and shell [125]. Currently, 16 clinically approved liposomal drugs are available, for example, DepoDur (morphine), AmBiosome (amphotericin B), DaunoXome (daunorubicin), Visudyne (verteporfin), DepoCyt (cytarabine), and Visudyne (verteporfin) [125]. All of these studies prove the efficacy of nanocarriers for the treatment of lung cancer and represent a promising strategy to improve the disease prognosis.

## Figures and Tables

**Figure 1 materials-13-05397-f001:**
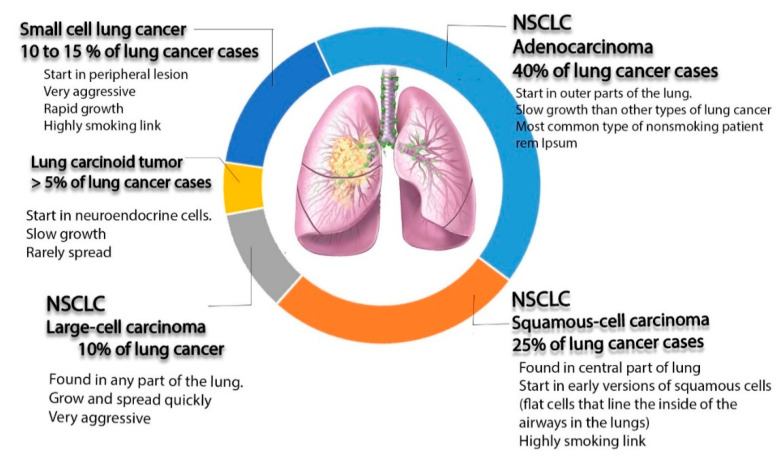
Schematic diagram illustrating different types of lung cancer (non-small cell lung cancer (NSCLC), small cell lung cancer (SCLC), and lung carcinoid tumors, as well as non-small cell lung cancer).

**Figure 2 materials-13-05397-f002:**
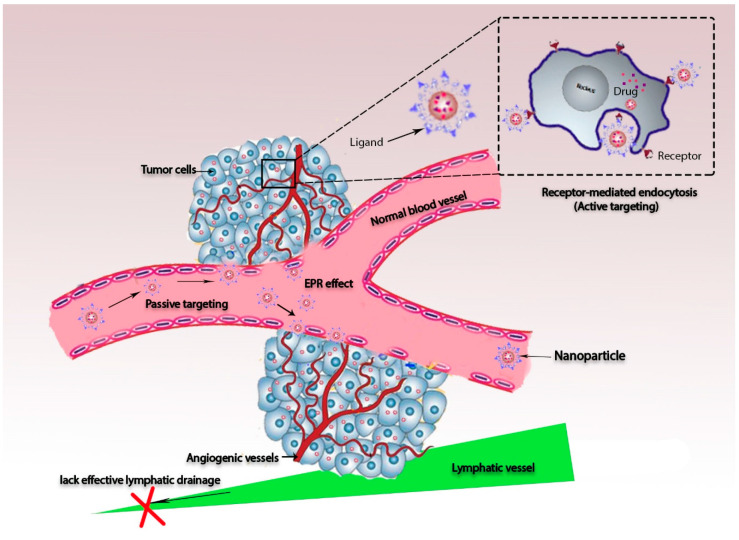
Schematic diagram of passive and active targeting strategies, showing accumulation of nanoparticles conjugated by ligands inside tumor tissue. The inset illustrates that the nanoparticle–ligand structure is recognized by receptors that are overexpressed on the cell membranes of cancer cells.

**Figure 3 materials-13-05397-f003:**
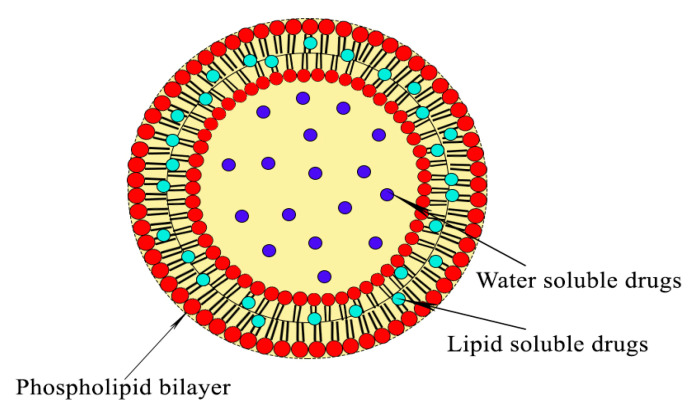
Structure of a liposome composed of a hydrophilic head exposed outwards and a tail that is turned into the bilayer as the hydrophobic stage. Water-soluble drugs can be inserted into the core of the liposome, while lipid soluble drugs are incorporated into the shell of the liposome.

**Figure 4 materials-13-05397-f004:**
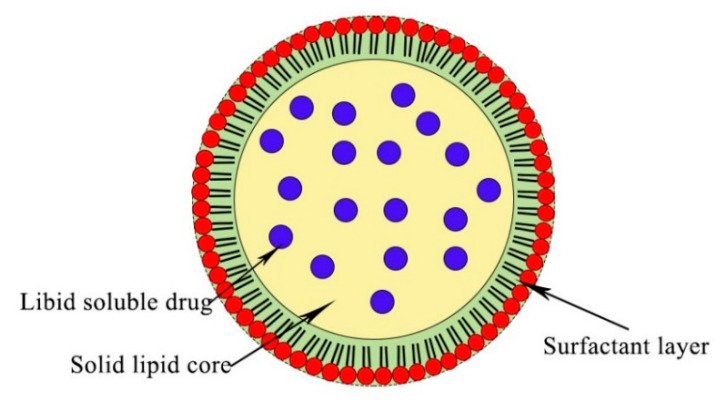
Structure of a solid lipid nanoparticle (SLNP) with lipophilic drugs inserted into its core. The drug molecule distribution in the lipid droplet depends on the use of appropriate techniques and the melting point for the solid lipid.

**Figure 5 materials-13-05397-f005:**
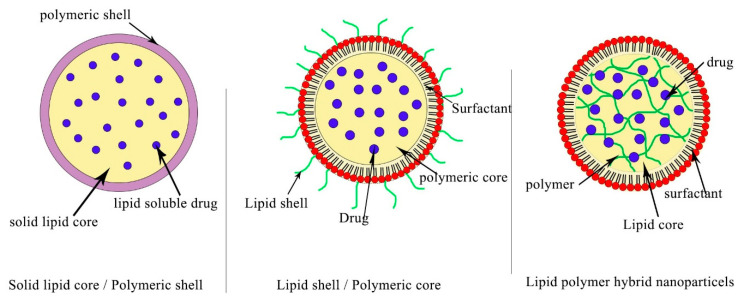
Polymeric NPs with several optimal design schemes. It was observed that the lipid core can be coated with polymeric materials, forming a lipid core and polymeric shell. In another case, polymeric materials can be condensed as a core template, and then a monolayer of lipids can be used to coat their surface, forming a face-like shape on the surface. Both the polymeric materials and lipid can be used to form the core template in the last case.

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
