# Peer review of "Nano targeted Therapies Made of Lipids and Polymers have Promising Strategy for the Treatment of Lung Cancer"

_materials, 2020, doi:10.3390/ma13235397_

Round 1
Reviewer 1 Report
This paper considers the nanoparticles based on polymers and lipid for treatment of lung cancer. The submitted manuscript is interesting, original and in the scope of the journal. In addition, the manuscript is well ilustrated, but some changes should be addressed:
- I didn't see in the introduction section the aim of the manuscript. Please highlight it.
- I didn't see in the manuscript where you refer to the table 1, 3 and 4.
- Please comment more on the information shown in tables (e.g. comparisons, own opinions).
- Please use "figure 3" instead "scheme 3" (line 147).
- Please consult the Instructions for Authors and arrange the references according to the guidelines (https://www.mdpi.com/journal/materials/instructions#preparation).
- At line 55 you have a reference without number (Curran et al.2001). Please correct it.
- All the references in the text must be modified, e.g. [1] instead of (1)
Author Response
Reviewer 1
This paper considers the nanoparticles based on polymers and lipid for treatment of lung cancer. The submitted manuscript is interesting, original and in the scope of the journal. In addition, the manuscript is well ilustrated, but some changes should be addressed:
I would like to thank reviewer 1 so much for his/her great efforts
- I didn't see in the introduction section the aim of the manuscript. Please highlight it.
Thank you so much for his/her good notification. Now aim of the review was added into introduction.
Here, we attempt to highlight the development and application of nanoparticles used in lung cancer therapies mainly those which were made of liposomes, lipid and polymer materials. Although many nanoparticle-based therapies have been developed for treatment of metastatic NSCLC. However, few have been translated successfully into clinical trials. In our current studies, liposome, lipid and hybrid polymeric nanoparticles were studied and they were summarized in table 1,2 and 3. While the successful nanoparticles that were moved into clinical trials were outlined in table 4.
- I didn't see in the manuscript where you refer to the table 1, 3 and 4.
Thank you so much for reviewer 1for his/her comment. Tables 1,3 and 4 are now written inside text.
- Please comment more on the information shown in tables (e.g. comparisons, own opinions).
Thank you so much for reviewer for his/her comment. In table 1, Tian and his team [62] designed hyaluronic Acid (HA)-modified mitochondria targeting smart liposomes with PTX to overcome multidrug resistance (MDR). The liposomes were prepared by thin-film hydration and then coated with hyaluronic acid (HA) by electrostatic adsorption, obtaining particle size about 100nm to ~140nm. The results administrated accumulation of nanoparticles in mitochnodria of tumor cells causing significant inhibition of cancer proliferation leading to overcome MDR.
Additionally, the dual targeting therapy was also developed by wang and his team using specific ligand peptide HAIYPRH (T7) and the cationic cell-penetrating peptide TAT that were connected with phospholipid via a polyethylene glycol (PEG) spacer to prepare the dual-ligand liposomes (T7/TAT-LP-PTX). This structure was evaluated on the tumor spheroids, which revealed that T7/TAT-LP was more efficaciously internalized in tumor cells than TAT-LP, T7-LP and LP, respectively [63]. Also, Almurshedi and his team [64], studied the preparation of cationic (CL) and pH-sensitive liposomes (PSL) for tyrosine kinase inhibitor afatinib (AFT) that was developed to enhance tumor-targetability against NSCLC cells in comparison to conventional liposomes (NL). The obtained liposomes showed particle size ˂100 nm with a spherical shape. The PSL, CL, and NL showed slow release profiles in pH 7.4. However, in acidic pH values, PSL exhibited fast release, which improved its tumor targetability. Moreover, AFT-loaded PSL inhibited the cell growth of human lung cancer cells (H-1975) more efficiently than free AFT, CL, and NL based on using Annexin V assay.
Zhu and his coworkers [67], investigated the efficacy of pulmonary delivery composed of docetaxel- folic acid-loaded liposomes (LPs-DTX-FA) as anti-cancer drugs in the form of nanoparticulate dry powders. It is suggested that the liposome-based inhaled dry powders represent a possibility of clinical therapy by achieving higher anti-cancer activity in tumor site and lower damage to healthy organs.
Tagami, and his team incorporated Poloxamer 188 (P188), into liposome-DOX membranes. The result suggested that DOX-loaded DPPC/P188 liposome may be useful for treating lung cancer. [69].
Zhang and his team developed poly(sialic acid) conjugated octadecylamine and then the mixture was attached liposomal pixantrone. The structure represents a promising approach for lung cancer treatment [70]. Many chemotherapeutics molecules were encapsulated inside moieties of liposomes to target lung cancer including Paclitaxel (PTX), Afatinib (AFT), Docetaxel (DTX) Erlotinib, Doxorubicin (DOX) and Pixantrone (PIX). These therapeutic molecules were summarized inside table 1.
Drugs that were encapsulated inside SLNPs were summarized in table 2. Since, Gupta et al developed solid lipid core nanocapsules (SLCN). This structure comprised a solid lipid core and a PEGylated polymeric corona and the paclitaxel (PTX) & erlotinib (ERL) were encapsulated inside this structure to be delivered into non-small cell lung cancer [80].
Kabary et al designed layer by layer alternate adsorption by using lactoferrin (LF) and hyaluronic acid (HA) upon surface of lipid nanoparticles (NPs) for dual delivery of berberine (BER) and rapamycin (RAP) to lung cancer. The obtained results showed highly recommendation to use LbL for enhancement functionalization of NPs and cellular uptake [81].
Khatri et al optimized Artemether loaded solid lipid nanoparticles to treat lung cancer. The results exhibited anti-lipolytic effect against lung cancer [82]
Rosière et al designed solid lipid nanoparticles coated by folate-grafted polyethylene glycol (PEG) and chitosan. Then paclitaxel was loaded SLN. The results showed positive impact of the coated SLN on the delivery of paclitaxel by inhalation [83]
Soni et al developed mannosylated solid lipid nanoparticles loaded by Gemcitabine using emulsification and solvent evaporation process. The result improved cellular uptake and drug efficacy of Gemcitabine inside lung cancer[84]
Shao, et al developed transferrin (Tf)-decorated nanostructured lipid carriers as multifunctional nanomedicine for co-delivery of paclitaxel (PTX) and enhanced green fluorescence protein plasmid [86].
Wang et al developed combination delivery of PTX and DOX using melt-emulsification technique of nanostructured lipid carriers. Results showed high cytotoxic effect among all formulations in vitro, as compared to single drug delivery [87].
The previous fabrication of polymeric nanoparticles were summarized inside table 3. Since Menon and his team developed afolate receptor nanoparticle containing a poly(N-isopropylacrylamide)-carboxymethyl chitosan shell and poly lactic-co-glycolic acid (PLGA) core. This assembly caused enhancement of localized chemo-radiotherapy to lung cancers [96]
Manadal and his team developed lipid monolayer coated biodegradable polymeric core (CSLPHNPs) for the delivery of erlotinib to treat non-small cell lung cancer. this assembly was produced by single-step sonication method using polycaprolactone (PCL) as the biodegradable polymeric core and phospholipid-shell composed of hydrogenated soy phosphatidylcholine (HSPC) and 1,2-distearoyl-sn-glycero-3-phosphoethanolamine-N-[methoxy(polyethylene glycol)-2000 (DSPE-PEG2000). The structure obtained particle size of about 170nm, PDI<0.2, drug entrapment efficiency of about 66% with good serum and storage stability [97].
Xiong and his coworkers designed combined therapy using cisplatin and metformin. Since cisplatin conjugated to polyglutamic acid to form anionic mixture, then the mixture reacted to cationic polymeric metformin. The payload was then inserted into liposome composed of DOTAP (2, 3-Dioleoyloxy-propyl)-trimethylammonium/Cholesterol/DSPE-PEG-anisamide aminoethyl.
Cisplatin loaded nanoparticles exhibited significantly increased tumor accumulation compared to free one without causing any nephrotoxicity [98].
Mottaghitalab, and her team optimized SP5-52 peptide conjugated silk fibroin nanoparticles loaded by gemcitabine to treat induced lung tumor in a mice model. the results showed significant reduction in lung tumor size compared with that of the untreated groups [99].
- Please use "figure 3" instead "scheme 3" (line 147).
Figure 3 was written instead of scheme 3
- Please consult the Instructions for Authors and arrange the references according to the guidelines (https://www.mdpi.com/journal/materials/instructions#preparation).
All references were arranged according to the journal style.
- At line 55 you have a reference without number (Curran et al.2001). Please correct it.
Curran et al.,2011 was corrected and now was given number [12]
- All the references in the text must be modified, e.g. [1] instead of (1)
All references in the text were modified

Reviewer 2 Report
Recommendation: Reject in the present form
General Comments:
The purpose of this review was to address how nanoparticles made of polymers and lipid (such us, liposomes, SNL nanoparticles and polymeric nanoparticles) have success to deliver an appropriate drug into lung cancer cells. The authors try to highlight those nanoparticles that have demonstrated more promising results and also the one that are already into clinical trials.
In my opinion, this review it is very poor. The references listed in this review are very scarce. In the last 2 decades there is been a “huge” amount of literature related on the development of “lipid nanocarriers” for cancer therapies purposes.
Another week point of this review, it is the lack of comparison between the three types of nanocarriers selected. I would expect to see this expressed in this review.
Specific points:
References
- The references cited in the manuscript are not in the right format of the journal. They should apperar [.] and not (.)
- The format of the references listed are not also in right format.
Figures:
The Fig. 3 (Scheme 3: Structure of liposome with two type of drugs inserted inside its core and shell) it is wrong.
As the authors mus know the liposome core it is water the drug inside should be “water soluble drug” and not “lipid soluble drug”. As the lipid bilayer it is hydrophobic the drug inside the lipid bilayer should be “lipid soluble drug” and not “water soluble drug”
Tables:
- All the tables are not well formated. The interior of the table should be separated by horizontal It is very difficult to follow.
Author Response
General Comments:
The purpose of this review was to address how nanoparticles made of polymers and lipid (such us, liposomes, SNL nanoparticles and polymeric nanoparticles) have success to deliver an appropriate drug into lung cancer cells. The authors try to highlight those nanoparticles that have demonstrated more promising results and also the one that are already into clinical trials.
In my opinion, this review it is very poor. The references listed in this review are very scarce. In the last 2 decades there is been a “huge” amount of literature related on the development of “lipid nanocarriers” for cancer therapies purposes.
I would like to thank reviewer 2 so much and I appreciate his/her great discussion and his/her comments.
Many new references were added as follows
- Lee, M.K. Liposomes for Enhanced Bioavailability of Water-Insoluble Drugs: In Vivo Evidence and Recent Approaches. Pharmaceutics. 2020, 13;12(3):264.
- Bozzuto, G.; Molinari, A. Liposomes as nanomedical devices. Int J Nanomedicine. 2015,10:975-999.
- Franzé, S.; Selmin, F.; Samaritani, E.; Minghetti, P.; Cilurzo, F. Lyophilization of Liposomal Formulations: Still Necessary, Still Challenging. Pharmaceutics. 2018,10(3):139.
- Arshinova, O.Y.; Sanarova, E.V.; Lantsova, A.V. et al. Lyophilization of liposomal drug forms (Review). Pharm Chem J.2012, 46, 228–233.
- Mukherjee, S.; Ray, S.; Thakur, R.S. Solid lipid nanoparticles: a modern formulation approach in drug delivery system. Indian J Pharm Sci. 2009,71(4):349-358
- Sarika,S.G.; Bhatt,L.K.Isotretinoin and α-tocopherol acetate-loaded solid lipid nanoparticle topical gel for the treatment of acne. Journal of Microencapsulation.2020,37:8, 557-565.
- Mishra, V.; Bansal, K.K.; Verma, A, et al. Solid Lipid Nanoparticles: Emerging Colloidal Nano Drug Delivery Systems. Pharmaceutics. 2018,10(4):191.
- Duan, Y.; Dhar, A.; Patel, C.; Khimani, M.; Neogi, S.; Sharma, P.; Siva Kumar, N.; Vekariya, R.L. A brief review on solid lipid nanoparticles: Part and parcel of contemporary drug delivery systems. RSC Adv.2020, 10, 26777–26791.
- Duong, V.A.; Nguyen, T.-T.-L.; Maeng, H.-J. Preparation of Solid Lipid Nanoparticles and Nanostructured Lipid Carriers for Drug Delivery and the Effects of Preparation Parameters of Solvent Injection Method. 2020, 25, 4781.
- Hanafy, N.A.N.; Fabregat, I.; Leporatti, S.; Kemary, M.E. Encapsulating TGF-β1 Inhibitory Peptides P17 and P144 as a Promising Strategy to Facilitate Their Dissolution and to Improve Their Functionalization. Pharmaceutics 2020, 12, 421.
- Hanafy, NA.; Dini, L.; Citti, C.; Cannazza, G.; Leporatti, S. Inhibition of Glycolysis by Using a Micro/Nano-Lipid Bromopyruvic Chitosan Carrier as a Promising Tool to Improve Treatment of Hepatocellular Carcinoma. Nanomaterials (Basel). 2018;8:1.
Another week point of this review, it is the lack of comparison between the three types of nanocarriers selected. I would expect to see this expressed in this review.
I would like to thank reviewer 2 and I appreciate his/her comment. Now the comparison between liposome, solid lipid nanoparticles and hybrid polymeric lipid nanoparticles was added.
As a comparison between liposome, solid lipid nanoparticles and hybrid polymeric lipid nanoparticles. It can be summarized that liposomes are mostly co formulated by presence of phospholipid and cholesterol forming core and shell in structure [101]. Their diameters depend mainly on methods of fabrication, surfactants, concentration of phospholipid and cholesterol, dehydration time and temperature. Architecture of liposome obtained much interest in encapsulating hydrophilic drug inside core and hydrophobic drug inside shell [102]. Hence, this assembly is characterized by encapsulating more than one type of drugs having different physicochemical properties. Additionally, the bilayer spherical system causes barrier to prevent leakage of drugs out of liposome moieties [103].
The disadvantage of liposomes is mostly attributed to their freeze drying technique [104] . Since, this technique leads to rupture their phospholipid membrane. To overcome this drawback, liposome is mostly coated by polymeric materials during fabrication such as PEGylated layer or they were incorporated inside CaCO3 -layer by layer technique.
While solid lipid nanoparticles are just solid core of lipid at 37°C formed by one of these techniques; high-pressure homogenization, double emulsion, high-shear homogenizer and emulsifier evaporation [105]. The lipid core can be then functionalized also by PEGylated layer or alternate adsorption [106] . Solid lipid nanoparticles are either bearing one drug incorporated inside core during fabrication or even more than one drug [107] . in general, the both hydrophobic and hydrophilic drugs can be inserted. The disadvantage of this structure is mostly associated to type of used lipid that can produce cytotoxicity related to free fatty acid and can be obtain growth of diameter during fabrication [108]. This mechanism translates their nanosized diameter into micro/ or macro sized [109]
Polymeric nanoparticles can be identified as interaction of two opposite polymeric materials forming cross linked network and condensed core [110] . This core can be further functionalized by PEGylated layer or it can be coated by lipid monolayer. Hanafy et al optimized hybrid polymeric lipid nanoparticles by using chitosan- oleic acid after blocking its free fatty acid [111]. This strategy reduced cytotoxicity that can be generated by oleic acid. Hybrid polymeric lipid nanoparticles can be optimized in shape of micelles, block co polymer or grafted co polymer. While drugs can be either inserted during fabrication or can be incorporated after fabrication. Recently, polymeric materials have been developed in shape of hydrogel materials, mucoadhesive materials and stimuli-responsive polymers.
Specific points:
References
- The references cited in the manuscript are not in the right format of the journal. They should appear [.] and not (.)
I would like to thank reviewer 2 so much for his/her great correction. All references are now marked by [.].
- The format of the references listed are not also in right format.
I would like to thank reviewer 2 so much for his great correction. All references are formatted in the journal style format.
Figures:
The Fig. 3 (Scheme 3: Structure of liposome with two type of drugs inserted inside its core and shell) it is wrong.
As the authors must know the liposome core it is water the drug inside should be “water soluble drug” and not “lipid soluble drug”. As the lipid bilayer it is hydrophobic the drug inside the lipid bilayer should be “lipid soluble drug” and not “water soluble drug”
I would like to thank reviewer 2 so much for his great correction and sorry for our mistake. Now scheme of liposome is corrected
Tables:
- All the tables are not well formatted. The interior of the table should be separated by horizontal It is very difficult to follow.
I would like to thank reviewer 2 The tables are managed now

Reviewer 3 Report
Authors have put sufficient efforts to make a review article titled "Nano targeted therapies made of lipid and polymers have a promising strategy for the treatment of lung cancer" Interesting. Although the content and strategy seem promising, it still needs improvement to arrange the article's flow.
Line no.74 Heading nanomedicine and cancer does not fit with the article as the article is about lung cancer. Line no. 75 explanation of nanotechnology at this point is not necessary. This whole subsection can be changed to show the nanoparticles in the target of lung cancer.
Line number 92 heading pulmonary drug delivery is again not the right title as per the review title. Heading 3 and 4 can be merged into one.
None of the figures has a description in it. An explanatory figure legend needs to be added to make it easy to understand for the reader.
The abstract is not giving enough information about the content of the review.
Too many tables, and there is no consistency in any of the tables.
Table 1 is not giving any specific information about the type of tumor. Table 2 has an extra column of advantage Table 3 has two more columns that are not consistent with the other two.
Table 4 should have two more columns about the results of completed trials and the assigned number.
References in the article are not consistent needs another check.
Author Response
Authors have put sufficient efforts to make a review article titled "Nano targeted therapies made of lipid and polymers have a promising strategy for the treatment of lung cancer" Interesting. Although the content and strategy seem promising, it still needs improvement to arrange the article's flow.
I would like to thank reviewer 3 for his/her great comments.
Line no.74 Heading nanomedicine and cancer does not fit with the article as the article is about lung cancer. Line no. 75 explanation of nanotechnology at this point is not necessary. This whole subsection can be changed to show the nanoparticles in the target of lung cancer.
Line number 92 heading pulmonary drug delivery is again not the right title as per the review title. Heading 3 and 4 can be merged into one.
I would like to thank reviewer 3 . Heading nanomedicine and cancer & heading pulmonary drug delivery were merged into one “Nanotechnology-mediated pulmonary drug delivery”
None of the figures has a description in it. An explanatory figure legend needs to be added to make it easy to understand for the reader.
I would like to thank reviewer 3 . description of figure was added to legend now
The abstract is not giving enough information about the content of the review.
I would like to thank reviewer 3. Abstract was improved as following;
The introduction of nanoparticles made of polymers, protein, and lipids as drug delivery system provides much progress in modern medicine. Since the application of nanoparticles in medicine means using biodegradable nanosized materials to deliver certain amount of chemotherapeutic agents into tumor site, leading to accumulate these nano encapsulated agents in right region. This strategy minimizes stress and toxicity generated by chemotherapies on healthy cells. Therefore, encapsulating chemotherapies have expressed their less cytotoxicity compared to non-encapsulation. The purpose of this review is to address how nanoparticles made of polymers and lipid can successfully be delivered into lung cancer. Thus, lung cancer types and their anatomies were introduced firstly to overview in general lung cancer structure. Then the rationale and strategy applied for using nanoparticles biotechnology in cancer therapies were discussed focusing on pulmonary drug delivery system made of liposomes, lipid nanoparticles and polymeric nanoparticles. Since many nanoparticles fabricated in shape of liposome, lipid nanoparticles and polymeric nanoparticles were summarized in table 1, 2 and 3 with focusing more on the encapsulated chemotherapeutic molecules, ligand-receptors attachment and their targets Afterwards, we have highlighted those nanoparticles that have demonstrated promising results and were delivered into clinical trials. The recent clinical trials that were done for successful nanoparticles were summarized in table 4.
Too many tables, and there is no consistency in any of the tables.
Table 1 is not giving any specific information about the type of tumor. Table 2 has an extra column of advantage Table 3 has two more columns that are not consistent with the other two.
I would like to thank reviewer 3 Tables are now consistency . Tables contain now the same heading column; Drugs, Composition, Preparation technique, and Target
In table 1 type of tumor was added , in table 2, advanatge column was re written . Table 3 was re arranged and improved
Table 4 should have two more columns about the results of completed trials and the assigned number.
I would like to thank reviewer 3. Table 4 was improved by adding column of assigned number. While, completed trial was presence in status column.
References in the article are not consistent needs another check.
I would like to thank reviewer 3. References were more re arranged

Round 2
Reviewer 1 Report
In my opinion, the introduction of this review is still poor. I recommend to improve this section by inserting a comparison between the lipidic and polymeric nanocarriers and other types of nanoparticles used as drug delivery systems (please see and use: doi: 10.3389/fmats.2020.00036).
Also, please try to improve section 2 (it has only 10 rows) or combine the section 2 with section 3.
Please revised the English language once again.
Author Response
In my opinion, the introduction of this review is still poor. I recommend to improve this section by inserting a comparison between the lipidic and polymeric nanocarriers and other types of nanoparticles used as drug delivery systems (please see and use: doi: 10.3389/fmats.2020.00036).
I would like to thank reviewer 1 so much for his/her great interest. I appreciate his/her correction.
The introduction is to overview the lung cancer types with description for their histological profile. So we would ask reviewer to give us permission to write the comparison in the line 362 as follow;
As a comparison between liposome, solid lipid nanoparticles and hybrid polymeric lipid nanoparticles, it can be summarized that liposomes are mostly co formulated by presence of phospholipid and cholesterol forming core and shell in structure [101]. Their diameters depend mainly on methods of fabrication, surfactants, concentration of phospholipid and cholesterol, dehydration time and temperature. Architecture of liposome obtained much interest in encapsulating hydrophilic drug inside core and hydrophobic drug inside shell [102]. Hence, this assembly is characterized by encapsulating more than one type of drugs having different physicochemical properties. Additionally, the bilayer spherical system causes barrier to prevent leakage of drugs out of liposome moieties [103].
On this way, liposomes possess great advantages as biodegradable and biocompatible, materials having large-scale production during fabrication allowing using them widely in biomedical applications. Besides that, entrapped drugs can be modulated inside the core and shell.
The disadvantage of liposomes is mostly attributed to possible oxidation of liposomal phospholipid layer leading to hydrolysis the membrane and degradation of liposomal structure. This drawback leads to leak drugs out of liposome. Furthermore, the lipid peroxidation leads to damage the properties of the lipid structure, particularly cellular permeability. Additionally, temperature, pH and light may induce instability for physical and chemical properties of liposomes leading to reduce the shelf life of liposomes during long storage. The other disadvantages could be related to use freeze- drying technique during liposomal fabrication [104]. Since, this technique leads to rupture their phospholipid membrane. To overcome those drawbacks liposomes are mostly coated by polymeric materials during fabrication such as PEGylated layer or they were incorporated inside CaCO3 by layer-by-layer technique.
While solid lipid nanoparticles are just solid core of lipid at 37°C formed by one of these techniques: high-pressure homogenization, double emulsion, high-shear homogenizer and emulsifier evaporation [105]. The lipid core can be then functionalized also by PEGylated layer or by alternate adsorption [106]. Solid lipid nanoparticles are either bearing one drug incorporated inside core during fabrication or even more than one drug [107]. In general, both hydrophobic and hydrophilic drugs can be inserted. For instance, solid lipid nanoparticles were characterized by their biocompatibility, low toxicity, good stability and enhancement of entrapped lipophilic drugs.
The disadvantage of this structure is mostly associated to type of used lipid that can produce cytotoxicity related to free fatty acid and can be obtain growth of diameter during fabrication [108]. Additionally, their diameter is growing up during fabrication to micro and macroparticles [109]
Polymeric nanoparticles can be identified as interaction of two opposite polymeric materials forming cross-linked network and condensed core [110]. This core can be further functionalised by PEGylated layer or it can be coated by lipid monolayer. Hanafy et al. optimized hybrid polymeric lipid nanoparticles by using chitosan- oleic acid after blocking its free fatty acid [111]. This strategy reduced cytotoxicity that can be generated by oleic acid. Hybrid polymeric lipid nanoparticles can be optimized in shape of micelles, block co polymer or grafted co polymer. While drugs can be either attached during fabrication or can be inserted after fabrication. Recently, polymeric materials have been developed in shape of hydrogel materials, mucoadhesive materials and stimuli-responsive polymers. For instance, polymeric nanoparticles characterized by controlling drug release, stable inside cells, and easy and cost effective formulation. However, the disadvantage of polymeric NPs is mostly associated with the type of used organic solvent during fabrication, and the polymer cytotoxicity.
Recently, advanced nanotechnology has obtained much progression in using materials for production and designing optimal drug delivery system. Since, the physicochemical properties of polymers used as advantage to develop both liposomes and solid lipid NPs and to overcome their drawbacks during fabrication. This new structure is called Polymeric hybrid lipid nanoparticles To Polymeric hybrid lipid nanoparticles contain three main structures as follows: (1) a hydrophobic/hydrophilic polymeric core inserted inside liposomes or coated by lipid monolayer; (2) Solid lipid core or liposomal phospholipid/ cholesterol layers were surrounded by polymeric shell; (3) polymer materials incorporated inside solid lipid or liposomal phospholipid/ cholesterol layers and then and an outer component, consisting of PEGylation layer was used.
Also, please try to improve section 2 (it has only 10 rows) or combine the section 2 with section 3.
I would like to thank reviewer1 for his /her comment. Section 2 and section 3 are now combined
Please revised the English language once again.
I would like to thank reviewer 1 . English is revised.

Reviewer 2 Report
General Comments:
The purpose of this review was to address how nanoparticles made of polymers and lipid (such us, liposomes, SNL nanoparticles and polymeric nanoparticles) have success to deliver an appropriate drug into lung cancer cells. The authors try to highlight those nanoparticles that try to show the more promising nanoparticles already into clinical trials. In spite of the changes, in my opinion, the revised version it is still poor. The authors improved the references and try to answer the questions, but unfortunately the English it is still very poor. The authors many times used “chemotherapies” instead “chemotherapeutic agents/or drugs” . The authors must improve the manuscript in order to be considered for publication in the Materials Journal.
Specific points:
The English of the Legends of all the figures and tables is very poor
Suggestion: Figure 1- Schematic diagram illustrating different types of lung cancers (non-small cell lung cancer (NSCLC), small cell lung cancer (SCLC) and lung carcinoid tumor and also non-small cell lung.
Suggestion: Figure 2. Schematic diagram of the passive and active targeting strategies showing accumulation of nanoparticles conjugated by ligand inside tumor tissue . The inset illustrates that nanoparticles-ligand were recognized by receptors that were overexpressed on cell membrane of cancer cells.
Suggestion :Figure 4: Structure of SLNP having lipophilic drugs inserted inside its core. The drug molecules distribution into the lipid droplet depends on the use of appropriate techniques and melting point for solid lipid.
Suggestion: Table 1 Nano-liposome encapsulating different types of drugs for treatment of lung cancer.
Suggestion: Table2 SLNPs encapsulating different types of drugs for treatment of lung cancer.
Suggestion: Table 3- Polymeric nanoparticles encapsulating different types of drug for treatment of lung cancer.
Suggestion: Table 4-Clinical trial studies in lung cancer therapies using lipid nanoparticles
line.74 -The authors wrote: Here, we attempt to highlight the development and application of nanoparticles used in lung cancer treatment mainly those which were made of liposomes, lipid and polymer materials. Although many nanoparticle-based therapies have been developed for treatment of metastatic NSCLC. However, few have been translated successfully into clinical trials. In our current studies, liposome, lipid and hybrid polymeric nanoparticles were studied and they were summarized in table 1,2 and 3. While the successful nanoparticles that were moved into clinical trials were outlined in table 4.
Q1. Are Liposomes not lipid materials? Do you mean Liposomes, Solid lipid nanoparticles (SLNP), Polymeric nanoparticles (PNPs) and Hybrid Polymeric materials?
Q2. The authors did not reply to my point. What are the advantages and disadvantages of the different type of nanoparticles used in lung cancer therapies?
In line 366 the authors wrote: The disadvantage of liposomes is mostly attributed to their freeze drying technique.
Q3. Are the authors sure about that? As the the authors must know there are another techniques to prepare liposomes! The freeze drying technique it is not the main technique used to produce liposomes.
Conclusions
The conclusions are very poor. I would expect that the author comments why the most of the nanoparticles in clinical trial are mainly liposomes?
References
Q5.References 74 and 122 are the same
74- Hanafy, N.A.N; El-Kemary, M.; Leporatti,,S. Micelles structure development as a strategy to improve smart cancer therapy, Cancers .2018, 10(7): 238.
122- Hanafy,N.A.N.; El-Kemary, M.; Leporatti,S. Micelles structure development as a strategy to improve drug delivery system. Cancer, 2018,10: 238
Author Response
The purpose of this review was to address how nanoparticles made of polymers and lipid (such us, liposomes, SNL nanoparticles and polymeric nanoparticles) have success to deliver an appropriate drug into lung cancer cells. The authors try to highlight those nanoparticles that try to show the more promising nanoparticles already into clinical trials. In spite of the changes, in my opinion, the revised version it is still poor.
The authors improved the references and try to answer the questions, but unfortunately the English it is still very poor. The authors many times used “chemotherapies” instead “chemotherapeutic agents/or drugs” . The authors must improve the manuscript in order to be considered for publication in the Materials Journal.
I would like to thank reviewer 2 so much for his/her comments. Chemotherapies were replaced by chemotherapeutic agents over all the text.
Specific points:
The English of the Legends of all the figures and tables is very poor
Suggestion: Figure 1- Schematic diagram illustrating different types of lung cancers (non-small cell lung cancer (NSCLC), small cell lung cancer (SCLC) and lung carcinoid tumor and also non-small cell lung.
Suggestion: Figure 2. Schematic diagram of the passive and active targeting strategies showing accumulation of nanoparticles conjugated by ligand inside tumor tissue . The inset illustrates that nanoparticles-ligand were recognized by receptors that were overexpressed on cell membrane of cancer cells.
Suggestion :Figure 4: Structure of SLNP having lipophilic drugs inserted inside its core. The drug molecules distribution into the lipid droplet depends on the use of appropriate techniques and melting point for solid lipid.
Suggestion: Table 1 Nano-liposome encapsulating different types of drugs for treatment of lung cancer.
Suggestion: Table2 SLNPs encapsulating different types of drugs for treatment of lung cancer.
Suggestion: Table 3- Polymeric nanoparticles encapsulating different types of drug for treatment of lung cancer.
Suggestion: Table 4-Clinical trial studies in lung cancer therapies using lipid nanoparticles
I would like to thank reviewer2 so much for his/her great interest. The English of the Legends of all the figures and tables is now corrected according to his/her suggestions.
line.74 -The authors wrote: Here, we attempt to highlight the development and application of nanoparticles used in lung cancer treatment mainly those which were made of liposomes, lipid and polymer materials. Although many nanoparticle-based therapies have been developed for treatment of metastatic NSCLC. However, few have been translated successfully into clinical trials. In our current studies, liposome, lipid and hybrid polymeric nanoparticles were studied and they were summarized in table 1,2 and 3. While the successful nanoparticles that were moved into clinical trials were outlined in table 4.
Q1. Are Liposomes not lipid materials? Do you mean Liposomes, Solid lipid nanoparticles (SLNP), Polymeric nanoparticles (PNPs) and Hybrid Polymeric materials?
I would like to thank reviewer 2. We mean Liposomes, Solid lipid nanoparticles (SLNP), Polymeric nanoparticles (PNPs) and Hybrid Polymeric materials
Q2. The authors did not reply to my point. What are the advantages and disadvantages of the different type of nanoparticles used in lung cancer therapies?
I would like to thank reviewer 2. Advantages and disadvantages of the Liposomes, Solid lipid nanoparticles (SLNP), Polymeric nanoparticles (PNPs) and Hybrid Polymeric materials were added as follows:
On this way, liposomes obtain great advantages as biodegradable and biocompatible, materials having large-scale production during fabrication allowing using them widely in biomedical applications. Besides that, entrapped drugs can be modulated inside the core and shell.
The disadvantage of liposomes is mostly attributed to possible oxidation of liposomal phospholipid layer leading to hydrolysis the membrane and degradation of liposomal structure. This drawback leads to leak drugs out of liposome. Furthermore, the lipid peroxidation leads to damage the properties of the lipid structure, particularly cellular permeability. Additionally, temperature, pH and light may induce instability for physical and chemical properties of liposomes leading to reduce the shelf life of liposomes during long storage. The other disadvantages could be related to use freeze- drying technique during liposomal fabrication [104]. Since, this technique leads to rupture their phospholipid membrane. To overcome those drawbacks, liposomes are mostly coated by polymeric materials during fabrication such as PEGylated layer or they were incorporated inside CaCO3 -layer by layer technique.
For instance, solid lipid nanoparticles were characterized by their biocompatibility, low toxicity, good stability and enhancement of entrapped lipophilic drugs.
The disadvantage of this structure is mostly associated to type of used lipid that can produce cytotoxicity related to free fatty acid and can be obtain growth of diameter during fabrication [108]. Additionally, their diameter is growing up during fabrication to form micro and macroparticles [109]
Polymeric nanoparticles are characterized by controlling drug release, stable inside cells, and easy and cost effective formulation. However, the disadvantage of polymeric NPs is mostly associated with the type of used organic solvent during fabrication, and the polymer cytotoxicity.
Recently, advanced nanotechnology has obtained much progression in using materials for production and designing optimal drug delivery system. Since, the physicochemical properties of polymers used as advantage to develop both liposomes and solid lipid NPs and to overcome their drawbacks during fabrication. This new structure is called Polymeric hybrid lipid nanoparticles To Polymeric hybrid lipid nanoparticles contain three main structures as follows: (1) a hydrophobic/hydrophilic polymeric core inserted inside liposomes or coated by lipid monolayer; (2) Solid lipid core or liposomal phospholipid/ cholesterol layers were surrounded by polymeric shell; (3) polymer materials incorporated inside solid lipid or liposomal phospholipid/ cholesterol layers and then and an outer component, consisting of PEGylation layer was used.
In line 366 the authors wrote: The disadvantage of liposomes is mostly attributed to their freeze-drying technique.
Q3. Are the authors sure about that? As the authors must know there are another techniques to prepare liposomes! The freeze-drying technique it is not the main technique used to produce liposomes.
I would like to thank reviewer 2 so much for this great comment. All the disadvantages were written beside the using of freeze-drying technique.
The disadvantage of liposomes is mostly attributed to possible oxidation of liposomal phospholipid layer leading to hydrolysis the membrane and degradation of liposomal structure. This drawback leads to leak drugs out of liposome. Furthermore, the lipid peroxidation leads to damage the properties of the lipid structure, particularly cellular permeability. Additionally, temperature, pH and light may induce instability for physical and chemical properties of liposomes leading to reduce the shelf life of liposomes during long storage. The other disadvantages could be related to use freeze- drying technique during liposomal fabrication [104]. Since, this technique leads to rupture their phospholipid membrane. To overcome those drawbacks, liposomes are mostly coated by polymeric materials during fabrication such as PEGylated layer or they were incorporated inside CaCO3 -layer by layer technique.
Conclusions
The conclusions are very poor. I would expect that the author comments why the most of the nanoparticles in clinical trial are mainly liposomes?
I would like to thank reviewer2. Conclusion is improved according to his/her suggestion
Nanotechnology has been used recently as an interesting approach to develop cancer therapies [118-124]. Since, functionalized nanoparticles have much interesting role in targeting of lung cancer cells by driving optimum dose into specific reaction using ligand–receptor conjugation. Liposome, solid lipid nanoparticles and polymeric based nanoparticles have the ability to overcome the respiratory tract barriers and mucociliary clearance to deliver drug into the deep part of lung. Although many different type of nanoparticles have been fabricated recently, however liposomes exhibit potential applications in biomedical filed and being derived into clinical trials because of their biodegradability, biocompatibility, their ability to produce large scale production, they can be functionalized by PEGylation or by Layer-by-Layer assembly allowing to prolong the circulation half-life. Additionally, they have small diameter, low toxicity, hybrid structure, ability to provide controlled and sustained release, and to modulate the distribution of drug inside core and shell [125]. To date, 16 clinically approved liposomal drugs are available, for example, DepoDur (morphine), AmBiosome (amphotericin B), DaunoXome (daunorubicin), Visudyne (verteporfin), DepoCyt (cytarabine), and Visudyne (verteporfin) 19]. All this studies prove the efficacy of nanocarriers in the lung cancer targeting, which is very promising strategy to improve the disease prognosis.
References
Q5.References 74 and 122 are the same
74- Hanafy, N.A.N; El-Kemary, M.; Leporatti,,S. Micelles structure development as a strategy to improve smart cancer therapy, Cancers .2018, 10(7): 238.
122- Hanafy,N.A.N.; El-Kemary, M.; Leporatti,S. Micelles structure development as a strategy to improve drug delivery system. Cancer, 2018,10: 238
Reference number 122 was replaced by this one
El-banna, F.S.; Mahfouz, M.E.; Leporatti , S.; El-Kemary, M.; and . Hanafy, N.A. Chitosan as a Natural Copolymer with Unique Properties for the Development of Hydrogels for the Development of Hydrogels. Appl. Sci. 2019, 9, 2193
